# Pediatric Tumors-Mediated Inhibitory Effect on NK Cells: The Case of Neuroblastoma and Wilms’ Tumors

**DOI:** 10.3390/cancers13102374

**Published:** 2021-05-14

**Authors:** Andrea Pelosi, Piera Filomena Fiore, Sabina Di Matteo, Irene Veneziani, Ignazio Caruana, Stefan Ebert, Enrico Munari, Lorenzo Moretta, Enrico Maggi, Bruno Azzarone

**Affiliations:** 1Immunology Research Area, Bambino Gesù Children’s Hospital, IRCCS, 00165 Rome, Italy; andrea.pelosi@opbg.net (A.P.); pierafilomena.fiore@opbg.net (P.F.F.); sabina.dimatteo@opbg.net (S.D.M.); irene.veneziani@opbg.net (I.V.); lorenzo.moretta@opbg.net (L.M.); 2Department of Paediatric Haematology, Oncology and Stem Cell Transplantation, University Children’s Hospital of Würzburg, 97080 Würzburg, Germany; Caruana_I@ukw.de (I.C.); ebert_s2@ukw.de (S.E.); 3Pathology Unit, Department of Molecular and Translational Medicine, University of Brescia, 25121 Brescia, Italy; enrico.munari@unibs.it

**Keywords:** neuroblastoma, Wilms’ tumor, NK cells, macrophages, tumor microenvironment

## Abstract

**Simple Summary:**

Neuroblastoma (NB) and Wilms’ tumor (WT) are the most common childhood solid extracranial tumors. The current treatments consist of a combination of surgery and chemotherapy or radiotherapy in high-risk patients. Such treatments are responsible for significant adverse events requiring long-term monitoring. Thus, a main challenge in NB and WT treatment is the development of novel therapeutic strategies to eliminate or minimize the adverse effects. The characterization of the immune environment could allow for the identification of new therapeutic targets. Herein, we described the interaction between these tumors and innate immune cells, in particular natural killer cells and monocytes. The detection of the immunosuppressive activity of specific NB and WT tumor cells on natural killer cells and on monocytes could offer novel cellular and molecular targets for an effective immunotherapy of NB and WT.

**Abstract:**

Natural killer (NK) cells play a key role in the control of cancer development, progression and metastatic dissemination. However, tumor cells develop an array of strategies capable of impairing the activation and function of the immune system, including NK cells. In this context, a major event is represented by the establishment of an immunosuppressive tumor microenvironment (TME) composed of stromal cells, myeloid-derived suppressor cells, tumor-associated macrophages, regulatory T cells and cancer cells themselves. The different immunoregulatory cells infiltrating the TME, through the release of several immunosuppressive molecules or by cell-to-cell interactions, cause an impairment of the recruitment of NK cells and other lymphocytes with effector functions. The different mechanisms by which stromal and tumor cells impair NK cell function have been particularly explored in adult solid tumors and, in less depth, investigated and discussed in a pediatric setting. In this review, we will compare pediatric and adult solid malignancies concerning the respective mechanisms of NK cell inhibition, highlighting novel key data in neuroblastoma and Wilms’ tumor, two of the most frequent pediatric extracranial solid tumors. Indeed, both tumors are characterized by the presence of stromal cells acting through the release of immunosuppressive molecules. In addition, specific tumor cell subsets inhibit NK cell cytotoxic function by cell-to-cell contact mechanisms likely controlled by the transcriptional coactivator TAZ. These findings could lead to a more performant diagnostic approach and to the development of novel immunotherapeutic strategies targeting the identified cellular and molecular targets.

## 1. Introduction

Natural killer (NK) cells play a major role in controlling tumor progression and metastatic spread by mediating tumor cell killing and producing pro-inflammatory cytokines and chemokines [1].

NK cells display a wide array of inhibitory and activating receptors on their cell surface. Normally, NK cell-killing is blocked by the binding of some inhibitor killer-cell immunoglobulin-like receptors (KIRs) with the major histocompatibility complex class I (MHC-I) molecules on healthy cells, but tumor cells generally lose MHC-I, thus becoming susceptible to NK-mediated killing. Moreover, ligands for the NK cell activating receptors are often upregulated or *de novo* expressed in tumor cells. The most important NK cell-activating receptors include natural cytotoxicity receptors (NCRs, NKp46, NKp30, and NKp44), DNAX accessory molecule-1 (DNAM-1), and NK group 2 member D (NKG2D), characterized in their transmembrane domains by sequences that, interacting with different adaptor proteins, trigger signals leading to the release of perforin and granzyme B, with consequent target cell lysis [2]. In addition, activated NK cells may also induce the apoptosis of tumor cells by releasing TNFα or through cell-to-cell contacts that activate the tumor necrosis factor (TNF)α–related apoptosis-inducing ligand (TRAIL) and FAS-ligand (FAS-L) pathways [1,3,4].

For these properties, harnessing NK cells derived from tumor patients or from healthy individuals has become major focus in tumor immunotherapy. For example, NK cells may be expanded ex vivo using various approaches in order to produce large amounts for adoptive transfer into cancer patients. However, adoptive cell transfer can be subverted by a patient’s immunosuppressive tumor microenvironment (TME), where cancer and stromal cells promote several immune evasion mechanisms targeting both innate and adaptive immune cells. These mechanisms have been widely explored in adult solid cancer, while in pediatric cancers, much fewer data are available.

Neuroblastoma (NB) and Wilms’ tumor (WT) are the two most common extracranial solid tumors in childhood whose prognosis remains unfavorable [5,6]. We will discuss in depth recent data revealing that, in both NB and WT, stromal and cancer cells display powerful inhibitory activity on NK cell function, leading to the impairment of their cytotoxic potential. In these pediatric tumors, the characterization of the immune environment and the impact on tumor cells could facilitate the identification of new therapeutic strategies capable of overcoming the inhibitory mechanisms, reducing toxicity and improving the long-term efficacy of the current treatments.

## 2. From Neuroblastoma to Wilms’ Tumor: The Impact on NK Cells

Several immunosuppressive cell subsets infiltrating the TME act through soluble factors [7,8,9,10,11,12,13]. Among these, cancer-associated fibroblasts (CAFs) can suppress NK cell function through the secretion of vascular endothelial growth factor (VEGF) by promoting the proliferation of regulatory T cells (Tregs) and the accumulation of myeloid-derived suppressor cells (MDSC) [13].

On the other hand, cancer cells themselves drive immunosuppressive activity on effector cells, including NK cells, releasing immunoregulatory molecules such as transforming growth factor β (TGF-β), prostaglandin E2 (PGE2), adenosine, lactate and kynurenine [14,15,16,17,18,19,20,21].

Furthermore, cell-to-cell contact represents another NK immunosuppressive mechanism operating in the TME. When NK cells were co-cultured with renal clear cell carcinoma (RCC) cells, the expression of the CD94/NK group 2 member A (NKG2A) inhibitory receptor on their surface was significantly increased and their cytolytic activity was reduced [22]. A number of studies have reported the existence of different soluble ligands of the NCR released by tumor cells which, upon interaction with NK cell receptors, may cause the blocking and/or modulation of NCR and its associated adaptor signaling molecules, with a consequent inhibition of cytotoxicity [23]. Accordingly, a combination of specific neutralizing monoclonal antibodies (mAbs) against NCR inhibits NK-cell-mediated tumor cell lysis in a more efficient way than the same mAbs used individually [24]. Importantly, both membrane-bound and soluble ligands can induce the down-regulation of NCR, while, in case of DNAM1, its inhibition is strictly dependent on cell-to-cell contact [24].

### 2.1. Immune Evasion Mechanisms of Neuroblastoma

NB is an embryonal tumor arising in tissues of the sympathetic nervous system, typically in the adrenal medulla or paraspinal ganglia. It is characterized by a biological heterogeneity resulting in a variety of clinical presentations, which may evolve either in total complete spontaneous regression (age < 1 year) or in the development of widespread metastatic tumors with very poor outcome [25]. Amplification of MYCN, a major oncogenic driver in neoplastic neuroblasts, patients’ age and the detection at diagnosis of a metastatic tumor identify NB as being at a high-risk of treatment failure [26]. At this stage, TME is described as cold or ‘immune-deserted’, since it is characterized by the presence of very few cytotoxic immune effector cells and by the expansion of regulatory T cells (Tregs) and MDSC. The latter cause the development of multiple immunomodulatory mechanisms, including MHC-I downregulation, which renders NB tumor cells “invisible” to cytolytic T lymphocytes [27,28].

By contrast, the detection of infiltrating immune effector cells is observed particularly in low-risk NB, where the presence of tumor-infiltrating lymphocytes (TILs) correlates with a favorable clinical outcome [29].

### 2.2. Neuroblastoma-NK Cell Interactions

In vitro data indicate that established neuroblastoma cell lines are highly susceptible to NK cell-mediated cell lysis due to the low expression of MHC-I and the presence of PVR, one ligand of the DNAM-1 activating receptor [30]. However, primary cultures from stage IV patients are less susceptible to NK cell lysis [31].

Ex vivo analysis strengthens the role of NK cells in the control of NB, since the combined infiltration of dendritic cells (DCs) and NK cells in NB-TME correlates with a favorable prognosis [32]. By contrast, NK cell function and recruitment are impaired in high-risk NB patients by the production of TGFβ and by the overexpression of B7-H3 molecules, which down-regulate NK cell cytotoxicity [33,34,35]. Differences between low-risk and high-risk NB may be explained either by the existence of two different tumor types or by an evolution occurring between the low- and high-risk NB, possibly reflecting changes in the TME. In the latter case, it would be important to understand the evolution occurring in the TME spanning from a situation of tumor protection in low-risk patients to the stage of “immune deserted TME” in high-risk patients. The emerging cell subsets inducing these changes have been not extensively investigated. In this context, no information was available, until recently, on the capacity of particular subsets of NB tumor cells or stromal cells to impair the immune response.

### 2.3. Neuroblastoma Stromal Cells

Different groups have recently identified, in primary human NB, a population of cancer associated fibroblasts (CAFs) that share phenotypic and functional characteristics with bone marrow-mesenchymal stem cells (BM-MSC) [36]. Notably, their transcriptomic profiling indicates that these CAF-MSCs were enriched in epithelial–mesenchymal transition genes in comparison with BM-MSCs [37]. Immunohistochemical analysis of human NB biopsies confirmed their presence inside the tumor stroma and their correlation with M2 tumor associated macrophages (TAM) [36]. CAF-MSCs display important tumor-promoting properties both in vitro and in vivo [36,38], and inhibit T cell functions much more efficiently than BM-MSC [37]. Since CAF-MSCs are present in the more aggressive NB [36], it is tempting to speculate that CAF-MSCs may be important players in the establishment of an immunosuppressive TME in high-risk NB, adapting their properties to the tumor stage. This has been reported, for instance, in pancreatic cancer, where the evolution and level of activation of stromal cells drive cancer aggressiveness [39]. Thus, in low-risk NB, TME could be composed of non-suppressive stromal cells while, in high-risk NB, an immunosuppressive stromal subset would favor tumor progression by impairing T, DCs and NK cell function. In this context, another putative stromal player is represented by Schwann cells, whose number in NB directly correlates with a favorable outcome and inversely correlates with a rich presence of CAFs [40]. Schwann cells display multiple immunomodulatory properties whose evolution may counteract or favor tumor progression [41,42].

### 2.4. Neuroblastoma Cancer Cells

NB is characterized by a non-random intratumoral heterogeneity controlled by super-enhancer-associated transcription factor (TF), which defines two divergent phenotypes: the committed adrenergic cells and an undifferentiated primitive mesenchymal subset identified by the expression of the CD133 marker [43]. Relapsing and metastatic NB are characterized by the enrichment of this undifferentiated mesenchymal subset that is more resistant to chemotherapeutic agents in vitro [43]. These cells acquire higher aggressiveness and potentially immunoregulatory function on T cells through the expression and activation of the transcriptional co-activator WW domain-containing transcription regulator-1 (WWTR1), also known as TAZ [43,44,45,46,47]. Indeed, TAZ and its paralog YAP are important effectors of the Hippo signaling pathway. These TF have pro-tumorigenic and immune-regulatory effects in multiple tumors [41,42,43,44], including NB, in which they have been involved in invasiveness and dissemination [48,49].

According to these findings, a cancer cell subset displaying a mesenchymal phenotype has been recently identified in NB cell lines, and thus termed neuroblastoma mesenchymal stromal cell (NB-MSC). These cells display potent inhibitory properties on NK cell function by cell-to-cell contact [50]. Notably, they expressed two potential immunoregulatory molecules: the ectoenzyme CD73 and the transcriptional coactivator TAZ. The extracellular adenosine generated by the ectonucleotidase CD73 is a newly recognized “immune checkpoint for cancer immunotherapy” that strongly inhibits anti-tumor immune responses [51]. TAZ displays important immunosuppressive effects [45,46,47], up-regulating the expression of programmed cell death protein 1-Ligand (PD-L1), the ligand for the checkpoint PD-1 expressed by both T and NK cells at the tumor site [44]. In addition, TAZ activation in NB has been reported to positively correlate with a worse prognosis [52]. Remarkably, the modulation of TAZ expression in NB-MSCs by RNA interference alters the expression of different immunosuppressive molecules such as platelet derived growth factor β (PDGFβ), PD-L1 and PD-L2 [50,53,54,55,56,57,58,59], affecting the cytotoxic activity of co-cultured NK cells.

Overall, these data highlight the unprecedented evidence that TAZ is able to control NK cell cytotoxic functions primarily thorough cell-to-cell contact mechanisms, which cause the downregulation of NCRs, DNAM1 and DAP12 adaptor signaling polypeptide [47], events that are sufficient to impair NK cell function. Contrary to what was previously reported in several NB cell lines [30], NB-MSCs cells displayed high levels of MHC-I and multifactorial resistance to NK cell-mediated lysis, including natural cytotoxicity and resistance to antibody-dependent cell-mediated cytotoxicity (ADCC) using both anti-CD105 IgG TRC105 and anti-GD2 mAb dinutuximab [50]. All these NB-MSC/NK cell interactions are detailed in Figure 1.

It is now clear that an efficient tumor eradication should contemporarily hit TME and cancer cells [60]. Therefore, even though the induction of ADCC in vivo with anti-GD2 and anti-CD105 mAbs offers a promising immunotherapeutic tool (since both markers are expressed by NB stromal and cancer cells) [36,37,38,61], this approach may be ineffective in the case of resistant NB-MSC-like cells [50]. In this context, it has been reported that the molecular mechanisms underlying the immune-suppressive activity of NB cells also involves the p53 onco-suppressor gene. Indeed, a poor susceptibility to the NK-cell-mediated killing of NB cell lines is associated with the presence of p53 mutations displayed by NB-MSC cells [62]. Moreover, Nutlin-3a, a compound that antagonizes the interaction between MDM2 and p53, may restore p53 transcription factor function, with a consequent increasing expression of ligands for NK cell-activating receptors on NB cells, enhancing the NK cell-mediated killing both in vitro and in vivo [63].

### 2.5. Diagnostic and Therapeutic Perspectives

An informative up-to-date diagnosis of NB will greatly benefit from the detection of novel criteria able to identify patients with either favorable or severe prognosis. In this context, as revealed by a recent paper, the detection of DCs and NK cells infiltrating the NB-TME positively correlated with both T-cell infiltration and a favorable clinical outcome. In agreement with these findings, two specific gene signatures related to DCs and NK cells were identified, showing that the expression of PD-1 and PD-L1 had a positive prognostic value and allowed for the prediction of the survival of NB patients, respectively [32]. On the other hand, the detection of CD133^+^TAZ^+^ cells [43], and that of the NB-MSC subset, allowed for the identification of high-risk patients for whom different immune therapeutic approaches should be explored. It would be also interesting to correlate the presence and the proportion of NB-MSCs with the development of “immune-deserted TME” [27]. From a therapeutic point of view, it would be important to eliminate or neutralize NB-MSCs and CD133^+^TAZ^+^ cells [44,64] by targeting, directly or indirectly, TAZ [46]. This could be achieved using Food and Drug Administration-approved drugs that indirectly block YAP/TAZ activation or critical downstream targets of YAP/TAZ, inducing the reduction of drug resistance in clinical trials [46]. For instance, the YAP/TAZ-TEAD inhibitor verteporfin could favor not only the elimination of immunosuppressive NB cell subsets but, more importantly, will induce the apoptosis of tumor initiating cells, sensitizing them to etoposide and cisplatin, the standard drugs used for NB therapy [65]. In addition, the elimination of CD133^+^TAZ^+^ cells could be obtained through the delivery of anti-miRNA using RNA nanoparticles targeting CD133, a treatment that, in preclinical models, induced the massive apoptosis of CD133^+^ cancer cells [66]. Finally, the in vivo removal of NB-MSCs could be achieved through the combination of anti-CD73 and immune checkpoint blockade, which has shown promising clinical results in patients with advanced adult solid tumors [51].

### 2.6. General Characteristics and Tumor Microenvironment of Wilms’ Tumor

WT, or nephroblastoma, is the most frequent renal tumor in children and has the second highest incidence, following leukemia, in patients aged less than five years. The overall survival for children with WT has reached 90% and 75% for localized and metastatic disease, respectively. However, WT patients with high-risk histology and/or tumor relapses have a survival rate of 50% [67,68].

WT is characterized by the presence of the nephrogenic rests derived from the residual embryonic renal cells resulting from the incomplete differentiation of metanephric blastema into the mature kidney [69]. Several germline and somatic genetic mutations linked to the control of fetal nephrogenesis characteristics are responsible for WT tumor histology and development [70,71]. WT development and progression has mainly been considered a genetic issue, while the existence and the role of TME in orchestrating an immunosuppressive response has been underestimated. In this context, a study has reported that the leukocyte infiltrate in WT is composed largely of macrophages in the necrotic areas of the tumor islands. A striking feature was represented by the rarity of DC, both within the tumor islands and in the peritumoral areas [72].

Immunohistochemistry studies in WT patients have revealed a robust expression of the inflammatory marker cyclooxygenase-2 (COX-2), strictly associated with the infiltration of TAM [73]. The presence of COX2 in TME resulted in the increased production of immunosuppressive cytokines such as IL-10 and TGF-β, and the expression of chemokine receptors such as C-C chemokine receptor type 5 (CCR5) and C-X-C motif chemokine receptor 4 (CXCR4), which, in turn, favored the infiltration of immunosuppressive cells such as plasmacytoid DC and Treg cells [74].

Moreover, the WT microenvironment is characterized by the presence of both tumor-infiltrating CD4 and CD8 T, mast cells and neutrophils, indicating that both adaptive and innate immune cells infiltrate WT [5,72,73,74]. The possible immunoregulatory role of different lymphoid subsets is further illustrated by the fact that the peripheral blood of children with WT is characterized by a significantly altered expression of Treg cells and NKT cells. This could influence the immune response and tumor development [75]. Therefore, the analysis of interactions between immune cells and WT cancer cells may be useful for identifying novel immunotherapies specifically targeting tumor cells and resulting in a decreased toxicity as compared to the current treatments.

### 2.7. WT Primary Cultures-NK Cell Interactions

#### 2.7.1. WT Components in Primary Cultures

A fundamental tool for investigating the mechanisms involved in WT/immune cell interactions is the availability of in vitro models reproducing such interactions. Up to now, the establishment of primary WT cultures expressing a stable phenotype has been difficult, and few cell lines expressing epithelial characteristics, but not blastemal or mesenchymal morphology, have been obtained [74,76,77,78,79]. Until recently, very few studies from different groups have reported the in vitro stabilization of WT primary cultures displaying mesenchymal [80,81], blastemal, or epithelial features [82,83,84,85]. Royer Pokora et al. stabilized in vitro primary cultures from the stromal component (str-WT), and showed that they bear the same mutations as the primary tumors. Gene expression profiling and phenotypic analysis revealed that these str-WT cells are similar to human MSCs; however, the WT cultures exhibit significant differences in their expression of transcription factors, thus indicating that a tumor-specific transcriptional program is activated in these cells [81]. Moreover, str-WT cells expressed some major ligands for activating and inhibiting NK cell receptors, as well as inhibitory checkpoint molecules involved in the negative regulation of anti-tumor immune response. str-WT exhibited potent inhibitory effects in vitro on the IL-2-induced proliferation of NK cells and on the expression of the activating receptors NKp30, NKp44, NKG2D, and 2B4. All these inhibitory effects were mediated by tumor-derived indoleamine-2,3-dioxygenase (IDO) metabolites and PGE2 [80].

#### 2.7.2. Blastemal and Epithelial Primary Cultures

Recently, our group reported the stabilization of four WT primary cultures expressing either a blastematous (CD56^+^/CD133^−^) or epithelial (CD56^−^/CD133^+^) phenotype, and investigated the effect of their interactions with NK cells and monocytes [86]. Data showed that both the blastemal and epithelial WT primary cultures expressed ligands for the NK activating receptors, namely DNAM-1 (CD112 and CD155) and NKG2D (MICA/B, and ULBP2–5-6). Accordingly, WT cells were efficiently killed by activated NK cells. Moreover, cytotoxicity was inhibited by the simultaneous use of neutralizing mAbs recognizing NCRs, DNAM-1 and NKG2D. However, both blastemal and epithelial WT primary cultures contained a cell fraction (~30%) resistant to NK-mediated cytotoxicity. These WT tumor cells, escaping NK-mediated killing, could represent a more aggressive subset. Notably, when co-cultured with NK cells, they were able to impair NK cell function. Remarkably, inhibition was not dependent on the release of soluble factors by WT cells, but could only be achieved upon cell-to-cell contact [86]. Thus, WT blastemal and epithelial primary cultures appeared to inhibit NK cell cytolytic function by a mechanism different from str-WT, highlighting the existence of a complex interplay in the WT TME aiming at NK cell anergy with different strategies. Moreover, NK cells, upon contact with WT primary cultures, displayed a markedly decreased expression of the checkpoint molecule T-cell immunoglobulin and the mucin-containing domain-3 (TIM-3). Interestingly, similar results were previously reported demonstrating that TIM-3 expression was downregulated on NK cells exposed to tumor targets, and this downregulation correlated with lower cytotoxicity and interferon gamma (IFN-γ) production [87]. In addition, T cell immunoglobulin and ITIM domain (TIGIT), strongly expressed in control NK cells, slightly decreased after co-culture with WT primary cultures and, in both situations, was associated with impaired NK cell cytolytic functions [86], as reported elsewhere [88]. Concerning the checkpoint inhibitor PD-1, it was strongly expressed in NK cells infiltrating several tumors [53,54,56,57,58]. Since WT is infiltrated by numerous NK cells [78,86] and it is also characterized by several PDL-1 positive tumor cells [86,89], it is likely that the in vivo blastematous and epithelial cells may lead to the impaired cytolytic activity of tumor-infiltrating NK cells. Interestingly, a previous paper reports opposite results, showing that the WT cell line HFWT greatly stimulates the proliferation of NK cells and of their PBMC precursors, leading to the robust expansion of NK cells able to kill fresh HFWT and K562 cells [90]. The use of a multi-passaged established cell line, differently to our data obtained using low passaged primary cells, could explain these discrepancies. On the other hand, we cannot rule out that, in low risk WT patients, tumor cells may differentially interact with NK cells, triggering a favorable immune response and limiting tumor expansion.

Finally, upon interaction with WT cells, monocytes became polarized towards alternatively activated M2 macrophages. In turn, M2 macrophages could further impair NK cell functions through cell-to-cell contact [86].

Remarkably, WT-driven M2 macrophages could represent a novel M2 subset, since they display the ability to inhibit degranulation acting, in addition, through cell-to-cell contact. Indeed, differently from what was reported, the IL-4- and MSC-induced M2 cells impair NK cell cytotoxicity through the secretion of IL-10 and TGFβ, which does not interfere with degranulation [91]. By contrast, these data are in agreement with a recent paper showing that murine peritoneal-, bone marrow-, and tumor-derived M2 cells inhibit NK cell lytic activity by cell-to-cell contact [92]. This involves TGFβ entrapped in the extracellular matrix or in a biologically active membrane-bound form [93,94].

In conclusion, WT stromal, blastemal and epithelial components can contribute both directly and indirectly to a broad immunosuppressive TME, which is likely to play a role in tumor progression. All of these WT/NK cell and WT/M2/NK cell interactions are detailed in Figure 2.

### 2.8. Diagnostic and Therapeutic Perspectives

For a long time, WT treatment strategies have been based on tumor staging and histology. Subsequently, a wide variety of biomarkers have been identified, and several of them could potentially allow for the better definition of more efficient treatment protocols [95]. Therefore, the current main aims of therapy are to prolong survival in patients with high-risk tumors and to decrease the side-effects of treatment. Accordingly, the identification of new specific targets is required to improve the clinical outcomes.

At the present, very few data are available on the presence (and frequencies) of NK cells infiltrating the blastemal and epithelial components of WT; thus, it is very difficult to predict if a high level of NK cell infiltration may constitute a favorable prognostic parameter or not.

Future immunohistological studies should assess the proportion of PD-1^+^ NK cells, both in the PB and within the tumor, in WT patients, and correlate these parameters with the clinical outcome. In addition, since the stromal component is highly inflamed and heavily infiltrated by macrophages [73,74], immunohistological studies should determine their type of polarization and correlate this feature with the clinical outcome.

In order to define a new strategy for determining immune evasion mechanisms exploited by WT blastemal or epithelial cells, our group queried public datasets to investigate if the parameter applied for NB setting could also be used on WT. This analysis revealed that both WT and renal cancers express high levels of TAZ as compared to non-pathologic tissue (Figure 3A). Indeed, it was found that higher levels of TAZ correlated with a worse overall survival in WT patients (Figure 3B). For this reason, for NB-MSCs, the targeting of TAZ could represent a possible treatment strategy [46]. Another useful strategy could be to interfere with the macrophage recruitment, macrophage survival, and macrophage polarization M2 [96], especially in association with combination therapies [97], using checkpoint inhibitors acting on NK cells [98,99,100,101,102]. For example, this could be achieved by associating the IL-15-based stimulation of NK cells with checkpoint inhibitors [98,99,101,102]. This strategy is feasible due to the high expression in WT primary cultures of ligands for both PD-1 and TIGIT.

## 3. Conclusions

The immunosuppression of NK cells represents an important pathogenic event occurring in the TME of adult and pediatric solid cancers. Here, we discussed emerging evidence showing that both tumor cells and different components of tumor stroma may participate in this process through several mechanisms, including signals mediated by cell-to-cell contact or by diffusible molecules. Notably, we described recent findings in NB and WT suggesting that cell contact between NK and cancer cells, rather than the release of diffusible molecules, may play a major role in NK cell immunosuppression in vitro. In addition, co-culture between WT primary culture and PB-monocytes induces, through both the release of soluble factors and cell-to-cell contact, the appearance of immunosuppressive M2 macrophages. Further work is needed to understand whether similar immuno-modulatory mechanisms also occur in vivo, and whether they can affect other potential anti-tumor effector cells, primarily T lymphocytes. Furthermore, in view of clinical translation, it is crucial to understand if these inhibitory effects could also hamper sophisticated cell-based immunotherapies, including chimeric antigen receptor (CAR) engineered NK and T cells [103,104,105,106].

According to the emerging role of YAP/TAZ in tumor immunity, our recent data support a model in which an aberrant TAZ up-regulation could confer potent immuno-modulatory properties on NK cells to cancer cells. However, the molecular pathways implicated in NK cell immunosuppression, and their clinical significance in these pediatric tumors, are far from being elucidated. The increased knowledge regarding the immunoregulatory properties of specific cancer cell subpopulations could pave the way for novel diagnostic and immunotherapeutic approaches to foster NK cell anti-tumor activity, with the potential for also broadening their application beyond pediatric solid tumors.

## Figures and Tables

**Figure 1 cancers-13-02374-f001:**
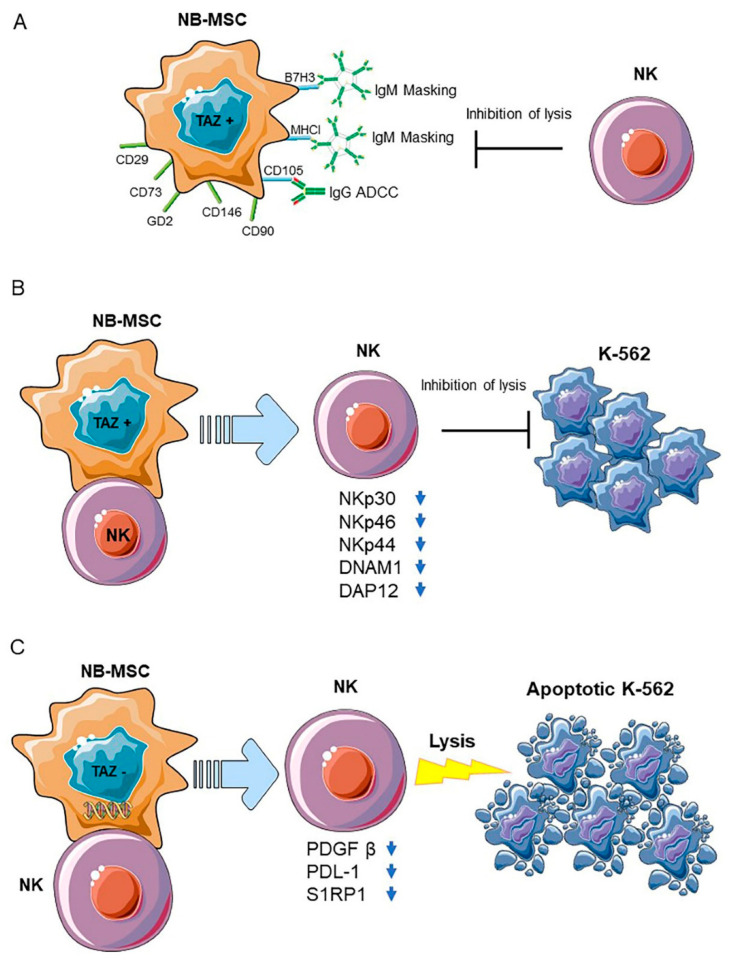
(**A**) NB-MSCs (CD105+/CD90+/CD73+/CD29+/CD146+/GD2+/TAZ+) exhibited multifactorial resistance to NK-mediated lysis. Activated NKs are unable to lyse NB-MSCs, even after B7-H3 or MHC-I masking or through ADCC mediated by anti-CD105 IgG and anti GD2 IgG. (**B**) NB-MSCs in contact with freshly isolated NK cells for 4 days induce a decreased capacity for killing K-562 cells in the latter through cell–cell contact-mediated mechanisms. This behavior is associated with a sharp decreased expression of natural cytotoxicity receptors (NCRs): NKp30, NKp46, and NKp44, of the adhesion and activating molecule DNAM1 and of the adaptor protein DAP12. (**C**) The latter property was partially controlled by TAZ, since its silencing in NB-MSC cells rescued freshly isolated NK-cell cytotoxic activity on the K-562 cells after 4 days of the coculturing of NB-MSC with NK cells efficiently. This is related to a reduction in the expression of platelet derived growth factor β (PDGF β), programmed death-ligand 1 (PDL-1), and sphingosine-1-phosphate receptor (S1PR1).

**Figure 2 cancers-13-02374-f002:**
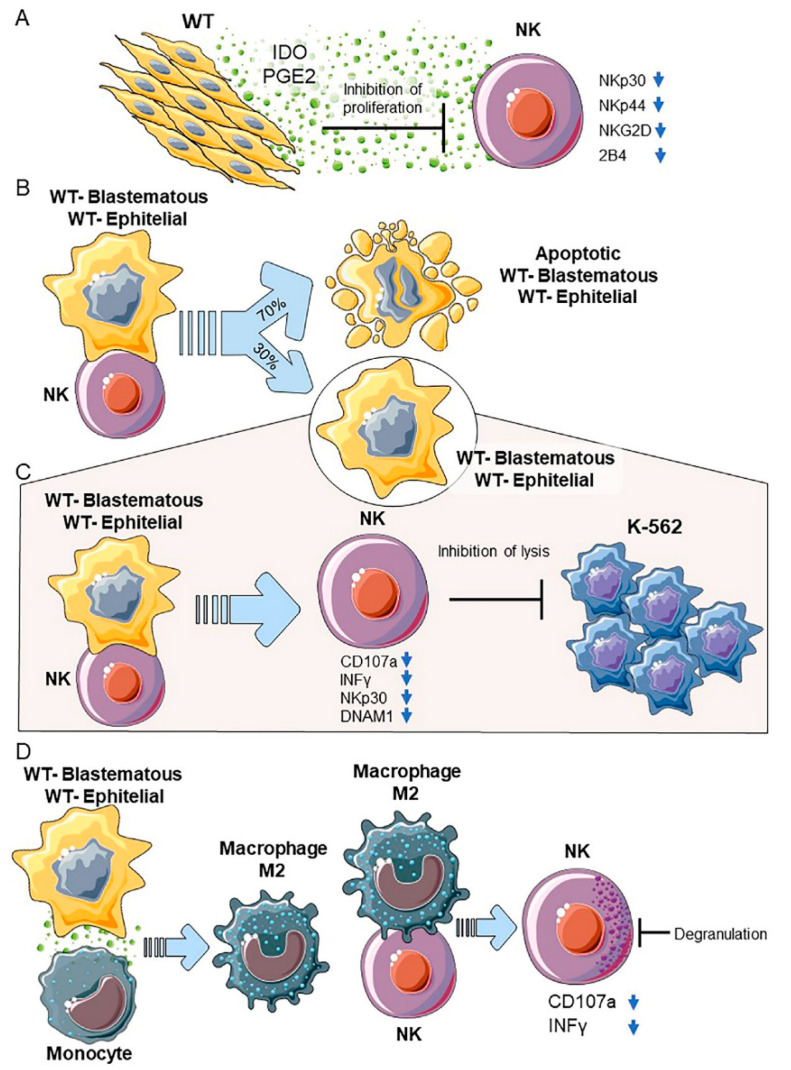
(**A**) WT-MSC cells inhibit the cytokine-induced proliferation of NK cells and the up-regulation of NKp30, NKp44, NKG2D and 2B4 through the secretion of the soluble immunoregulatory factors IDO and PGE2. (**B**) Efficient NK-mediated lysis (70%) of WT-blastemal and WT-epithelial cells. (**C**) 30% surviving WT-blastemal and WT-epithelial cells after 4 days of coculture, efficiently inhibiting the lytic function of activated NK cells against the K-562 target cells. (**D**) WT-blastemal and WT–epithelial after 6 days of coculture with PB-monocytes, inducing their polarization into M2 macrophages. M2 macrophages after 3 days of coculture with freshly isolated NKs inhibit their production of CD107a and IFN-γ.

**Figure 3 cancers-13-02374-f003:**
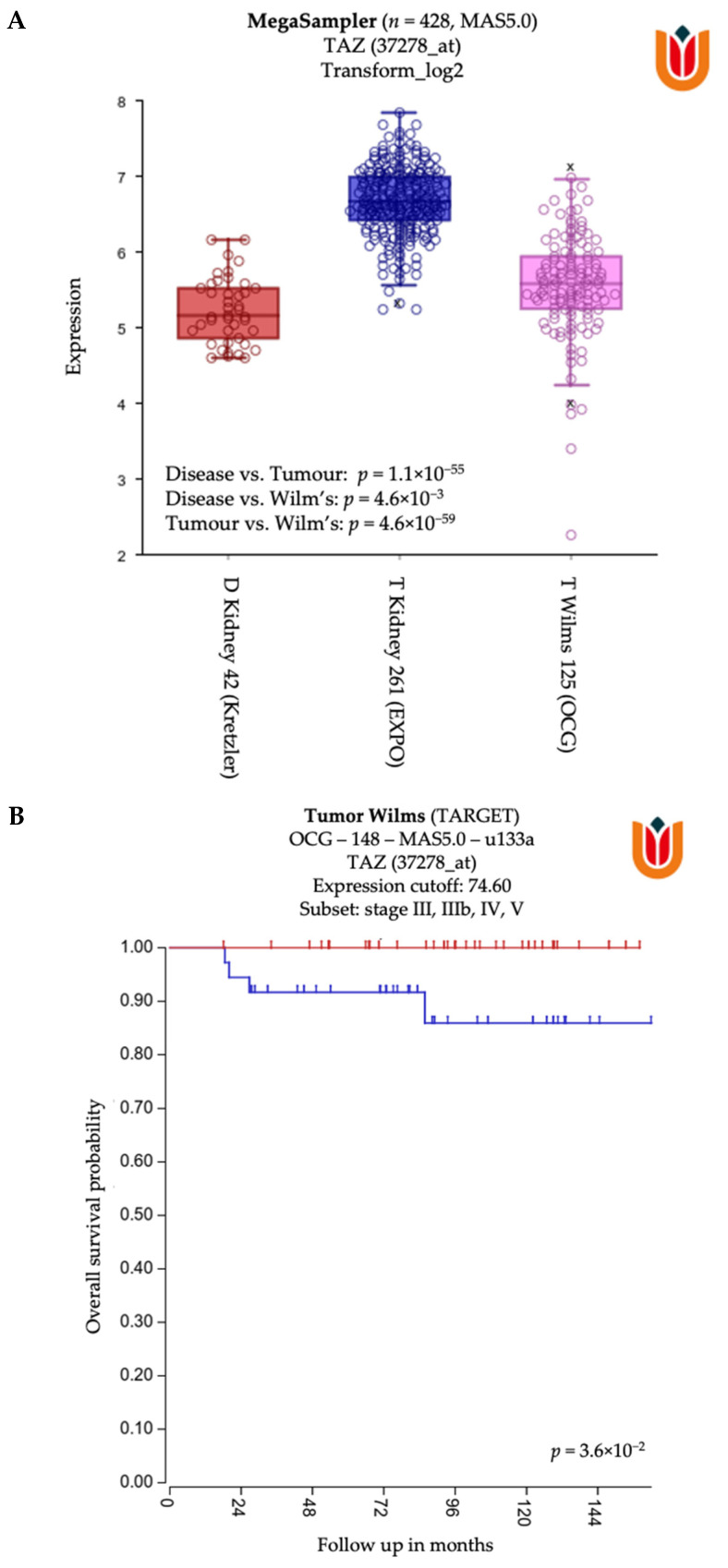
(**A**). “Megasampler” analysis across R2 public dataset (https://hgserver1.amc.nl/, accessed on 31 March 2021) revealed that both WT and renal cancers express high levels of TAZ compared to normal non-pathological tissue. (**B**) A further analysis on WT dataset (TARGET-OCG-148-MAS5.0-u133a) underlined that, in WT patients, higher levels of TAZ correlated with a worse overall survival.

## Data Availability

Analysis in Figure 3 are generated from R2 public datasets (https://hgserver1.amc.nl/, accessed on 31 March 2021).

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
