# Peer review of "Pediatric Tumors-Mediated Inhibitory Effect on NK Cells: The Case of Neuroblastoma and Wilms’ Tumors"

_cancers, 2021, doi:10.3390/cancers13102374_

Round 1

Reviewer 1 Report

This is an interesting review. However, I am  afraid that I find the presentation tedious, cumbersome, and long-winded in many places. A continuous narrative is not always obvious. The data could be better synthesised and presented in a more concise manner.

This particularly illustrated by passages that appear confusing and self-contradictory, such as the paragraph "Neuroblastoma-NK cell interactions". Here, the findings are presented in a largely unfiltered way, more or less as a list and not in a clear structure that explains the findings in a coherent context. Similarly, information appears randomly assembled throughout the whole manuscript (although it is upon more intensive engagement clear that it is not).

Also, empty sentences that do not contain concrete information could be avoided, e.g. "Innate immune responses are involved in tumor immune surveillance."

Generally, the manuscript is wordy and often repetitive. Often the same information is repeated in slightly different words, e.g.:

"Regardless of the type of solid tumor, the stroma represents an important component of the TME. It is normally composed by heterogeneous cell types which in most cases acquire an immunosuppressive phenotype and they inhibit immune cells with effector function including NK cells by a plethora of immunomodulatory mechanisms. Tumor-associated mesenchymal stromal cells (TA-MSCs), cancer-associated fibroblasts (CAF) or activated stellate cells derived from patients with different solid tumors mediate inhibition of NK cell function through releasing of immunosuppressive factors such as prostaglandin E2 (PGE2), Indoleamine 2,3-dioxygenase (IDO), tumor growth factor β (TGFor the chemokine CXCL12 [8-14]. In addition, CAFs can suppress NK cell function through secretion of VEGF, by promoting the proliferation of regulatory T cells (Tregs) and the accumulation of myeloid-derived suppressor cells (MDSC) [14]." The first two sentence are not needed. 

Although the focus should primarily lie on the quality of the information and not of the language, critical proofreading by someone with profound English skill might be helpful.

Further examples:

  • The use of many abbreviations does not support the understanding.
  • lines 32/33: "Recently, in both tumor entities have been demonstrated the presence of subsets of tumor cells with a strong inhibititory capacity against NK cells with mechanisms different from those decribed in adult cancers." Sentences like this could be broken down into shorter ones to make them easier intelligible.
  • lines 33/34: "These recent acquisitions concerning the immunoregulatory properties of 34 NB and WT cells could facilitate the identification of cellular and molecular targets leading to the 35 development of novel and more efficient immunotherapeutic strategies." This sentence is so vague that it does not provide tangible information.
  • lines 104/105: "It is characterized by different molecular abnormalities at genomic, epigenomic and transcriptomic levels [25-27]." This is another example of a very vague sentence.

Author Response

Point-by point reply to Reviewer N°1

We want to thank the reviewer for his incisive comments that we have followed trying to make the text smoother and more readable.

General comments

This is an interesting review. However, I am  afraid that I find the presentation tedious, cumbersome, and long-winded in many places. A continuous narrative is not always obvious. The data could be better synthesised and presented in a more concise manner.

This particularly illustrated by passages that appear confusing and self-contradictory, such as the paragraph "Neuroblastoma-NK cell interactions". Here, the findings are presented in a largely unfiltered way, more or less as a list and not in a clear structure that explains the findings in a coherent context. Similarly, information appears randomly assembled throughout the whole manuscript (although it is upon more intensive engagement clear that it is not).

We thank you the reviewer for this comment that we appreciate. Following his suggestions we have shortened the manuscript trying to avoid repetitions and empty sentences. Thus, we have rewritten and reorganized the sections criticized by the reviewer, as well as also the other sections.

Specific comments

Also, empty sentences that do not contain concrete information could be avoided, e.g. "Innate immune responses are involved in tumor immune surveillance."

We have deleted this sentence.

Generally, the manuscript is wordy and often repetitive. Often the same information is repeated in slightly different words, e.g.:

"Regardless of the type of solid tumor, the stroma represents an important component of the TME. It is normally composed by heterogeneous cell types which in most cases acquire an immunosuppressive phenotype and they inhibit immune cells with effector function including NK cells by a plethora of immunomodulatory mechanisms. Tumor-associated mesenchymal stromal cells (TA-MSCs), cancer-associated fibroblasts (CAF) or activated stellate cells derived from patients with different solid tumors mediate inhibition of NK cell function through releasing of immunosuppressive factors such as prostaglandin E2 (PGE2), Indoleamine 2,3-dioxygenase (IDO), tumor growth factor β (TGFor the chemokine CXCL12 [8-14]. In addition, CAFs can suppress NK cell function through secretion of VEGF, by promoting the proliferation of regulatory T cells (Tregs) and the accumulation of myeloid-derived suppressor cells (MDSC) [14]." The first two sentence are not needed

We have deleted the two sentences and rephrased the paragraph.

Although the focus should primarily lie on the quality of the information and not of the language, critical proofreading by someone with profound English skill might be helpful.

We have performed the requested proofreeding.

Further examples:

    • The use of many abbreviations does not support the understanding.
    •  
  • For each abbreviation we have added the whole definition and we have tried to limit their employ.
    •  
  • lines 32/33: "Recently, in both tumor entities have been demonstrated the presence of subsets of tumor cells with a strong inhibititory capacity against NK cells with mechanisms different from those decribed in adult cancers." Sentences like this could be broken down into shorter ones to make them easier intelligible.
    • As suggested , we have broken down this sentence in shorter ones.
    • lines 33/34: "These recent acquisitions concerning the immunoregulatory properties of 34 NB and WT cells could facilitate the identification of cellular and molecular targets leading to the 35 development of novel and more efficient immunotherapeutic strategies." This sentence is so vague that it does not provide tangible information.
  •  
  • We agree with this comment, and we have modified this sentence trying to offer a more concrete information on the cellular and molecular targets.
  • Thus, following the above mentioned reviewer’s comments we have rephrased the last part of the summary.
    •  
  • lines 104/105: "It is characterized by different molecular abnormalities at genomic, epigenomic and transcriptomic levels [25-27]." This is another example of a very vague sentence.
  •  
  • We have deleted this sentence and modified the beginning of the introduction.
  •  

Reviewer 2 Report

In the manuscript, Pelosi and colleagues aimed to review mechanisms of impairing the activation and function of natural killer (NK) cells focusing on the most frequent pediatric extra cranial tumors, Neuroblastoma (NB) and Wilms’ tumor (WT). More particularly, authors report on mechanisms impacting on an immunosuppressive tumor microenvironment (TME) including myeloid-derived suppressor cells (MDSC), tumor-associated macrophages (TAM), regulatory T cells (Tregs) and malignant cells and their direct cell to cell or indirect interrelationship with NK cells. In addition, they report on identification of cellular and molecular targets that may pave the way to the development of novel efficient immunotherapeutic strategies in these childhood cancers.

In summary, this is a well performed and informative compendium on basic and functional properties of NK cell activity in pediatric malignancies that is further supported by three clearly arranged figures. The review comprises an extensive compilation of the most relevant references in the field and may help the reader to get easy access to current concepts and putative therapeutic relevance of NK cells. There are no major issues regarding this manuscript as the topic addressed is of scientific relevance and authors put high efforts in preparing the manuscript.  There are, however, some minor points as mentioned successive aiming to further improve the readability and significance of the manuscript.

Minor points of improvement:

  1. The review may further benefit from including a more detailed description of the biology and functional properties of NK cell receptors NKp30, NKp46, NKp44, NKG2D and DNAM1.
  2. Line 77: “ …tumor growth factor” may be “transforming growth factor beta”.
  3. Authors mostly introduced acronyms at their first appearance in the text but did not consequently performed that throughout the entire text. Thus, some additional abbreviations (e.g. TRAIL, FAS-L, VEGF, DNAM1 …) should carefully be introduced. In line with that, acronyms depicted in the figures should be given in the legends to the figures.

Author Response

Point-by point reply to Reviewer N°2

We want to thank the reviewer for his appreciation for our manuscriipt and for his helpful comments.

Minor points of improvement:

  1. The review may further benefit from including a more detailed description of the biology and functional properties of NK cell receptors NKp30, NKp46, NKp44, NKG2D and DNAM1.

We have added the requested information concerning the NK cell activatory receptors as second paragraph of the introduction.

  1. Line 77: “ …tumor growth factor” may be “transforming growth factor beta”.

We have corrected this point.

  1. Authors mostly introduced acronyms at their first appearance in the text but did not consequently performed that throughout the entire text. Thus, some additional abbreviations (e.g. TRAIL, FAS-L, VEGF, DNAM1 …) should carefully be introduced. In line with that, acronyms depicted in the figures should be given in the legends to the figures.

We are sorry for this misunderstanding, we have introduced each abbreviation with the appropriate acronym.

Reviewer 3 Report

The review article “Pediatric tumors-mediated inhibitory effect on NK cells: the case of Neuroblastoma and Wilms’ tumors” by Pelosi et al, tried to review mechanisms of NK cell inhibition in neuroblastoma and Wilms tumor. The review is highly unfocused and lacks clarity. There are several reviews available highlighting the importance of NK cells and how their activity is diminished in neuroblastoma (PMIDs: 31502008, 24575100, 32457760). The current review doesn’t add any new information to the existing literature available. The role of NK cells in Wilms tumor and mechanisms of their recruitment and activation in Wilms tumor might be of interest to the readers, but authors have not provided updated literature on Wilms tumor.

Author Response

Point by point reply to Reviewer N°3

We respect, but we desagree with the negative comments of this reviewer, especially concerning the points 2 and 3.

The review article “Pediatric tumors-mediated inhibitory effect on NK cells: the case of Neuroblastoma and Wilms’ tumors” by Pelosi et al, tried to review mechanisms of NK cell inhibition in neuroblastoma and Wilms tumor. The review is highly unfocused and lacks clarity.

We have shortened the manuscript and we have rewritten and reorganized the whole text in order to make it smoother, more readable and more focused.

There are several reviews available highlighting the importance of NK cells and how their activity is diminished in neuroblastoma (PMIDs: 31502008, 24575100, 32457760). The current review doesn’t add any new information to the existing literature available.

Indeed, we reported recent data showing, for the first time, the existence of a novel tumor cell subset (NB-MSC) displaying i) multifactorial resistance to NK cell-mediated killing. ii) strong inhibition of NK cytotoxic function and iii) the role of TAZ in the mechanism of inhibition of NK function.

The role of NK cells in Wilms tumor and mechanisms of their recruitment and activation in Wilms tumor might be of interest to the readers, but authors have not provided updated literature on Wilms tumor.

We have cited literature spanning from 1993 to 2021 (about 20 papers) covering all the key aspects concerning Wilms Tumor! In addition, we have also added two recent papers concerning infiltrating CD8+T cells and TGFb as possible biomarkers of WT. Therefore, we believe to have provided updated literature.

Round 2

Reviewer 3 Report

The authors have tried to reorganize the manuscript to make it more focused. However, the manuscript needs to be revised before acceptance. 

  • In the introduction, the authors have added info about activating receptors of NK cells which is highly elaborated and lacks clarity. Rephrase it and just mention the names of the activating receptors.
  • Line 237: Discuss in detail the therapeutic potential of targeting TAZ in neuroblastoma, including the potential use of TAZ inhibitors available. Add current literature highlighting targeting TAZ in NB (PMID 33347823).
  • Add references in heading  TME of Wilms tumor (27974111, 15246157)

      4) Line 382: Add references related to modulating M2 polarization strategies (PMID: 32983125, 28117416)

Author Response

The authors have tried to reorganize the manuscript to make it more focused. However, the manuscript needs to be revised before acceptance. 

We thank the reviewer for his/her helpful suggestions. All the modified sentences are in bold and in blue characters.

1) In the introduction, the authors have added info about activating receptors of NK cells which is highly elaborated and lacks clarity. Rephrase it and just mention the names of the activating receptors.

As suggested, in the Introduction we have rephrased and shortened the third sentence concerning the description of the NK activatory receptors. We had added this sentence according to the suggestion of Reviewer 2 who asked for this paragraph. Therefore, we have tried to maintain a balance between the comments of the two We hope that the revision of this paragraph will satisfy both reviewers.

2) Line 237: Discuss in detail the therapeutic potential of targeting TAZ in neuroblastoma, including the potential use of TAZ inhibitors available. Add current literature highlighting targeting TAZ in NB (PMID 33347823).

We thank the reviewer for the suggested reference that is very interesting and important concerning TAZ pharmacological targeting. As proposed, we have discussed more in detail the therapeutic potential of targeting TAZ in neuroblastoma citing the proposed reference that shows that the YAP/TAZ-TEAD inhibitor Verteporfin causes the elimination of the NB Tumor Initiating Cells. Our comment has been included in the paragraph Diagnostic and therapeutic perspectives of the NB section.

3) Add references in heading  TME of Wilms tumor (PMID: 27974111, PMID: 15246157)

Following your suggestion we have cited reference (PMID: 27974111) adding a sentence of comment at the end of the heading General characteristics and tumor microenvironment of Wilms’ Tumor.

Concerning reference (PMID: 15246157), we preferred to add and comment this reference with a sentence that has been included in the heading Blastemal and epithelial primary cultures, at the end of the paragraph dedicated to WT primary cultures/NK cells interactions. Since the above-mentioned reference reports that  the co-culture of the  established WT cell line HFWT  with NK cells induces effects on their proliferation  and lytic function that are different from those that we have described, we thought that it was preferable comment the reference in this section of the manuscript.

 4) Line 382: Add references related to modulating M2 polarization strategies (PMID: 32983125, PMID: 28117416)

Thank you for this suggestion. We have added the references, slightly modifying the last sentence of the heading Diagnostic and therapeutic perspectives.